Corrected: Publisher correction

# Development of a genetically encodable FRET system using fluorescent RNA aptamers

Mette D.E. Jepsen [1], Steffen M. Sparvath [1], Thorbjørn B. Nielsen [1,2], Ane H. Langvad [1,3], Guido Grossi [1], Kurt V. Gothelf [1,2] & Ebbe S. Andersen [1,3]

Fluorescent RNA aptamers are useful as markers for tracking RNA molecules inside cells and for creating biosensor devices. Förster resonance energy transfer (FRET) based on fluorescent proteins has been used to detect conformational changes, however, such FRET devices have not yet been produced using fluorescent RNA aptamers. Here we develop an RNA aptamer-based FRET (apta-FRET) system using single-stranded RNA origami scaffolds. To obtain FRET, the fluorescent aptamers Spinach and Mango are placed in close proximity on the RNA scaffolds and a new fluorophore is synthesized to increase spectral overlap. RNA devices that respond to conformational changes are developed, and finally, apta-FRET constructs are expressed in *E. coli* where FRET is observed, demonstrating that the apta-FRET system is genetically encodable and that the RNA nanostructures fold correctly in bacteria. We anticipate that the RNA apta-FRET system could have applications as ratiometric sensors for real-time studies in cell and synthetic biology.

[1] Interdisciplinary Nanoscience Center, Aarhus University, 8000 Aarhus C, Denmark. [2] Department of Chemistry, Aarhus University, 8000 Aarhus C, Denmark. [3] Department of Molecular Biology and Genetics, Aarhus University, 8000 Aarhus C, Denmark. Mette D.E. Jepsen and Steffen M. Sparvath contributed equally to this work. Correspondence and requests for materials should be addressed to E.S.A. (email: esa@inano.au.dk)

Green fluorescent protein (GFP)[1, 2], has, since its development in the 1990s, been widely used to label and track proteins in cellular environments. Analogously to fluorescent proteins (FPs), tracking of RNA in cells has been achieved using fluorescent RNA aptamers, which are structured RNA molecules that bind and enhance the fluorescence of small-molecule fluorophores[3]. The first demonstration of a fluorescent RNA aptamer was the malachite green aptamer[4], which upon binding of malachite green increased its fluorescence more than 2000-fold. Malachite green is however toxic to yeast and mammalian cells[5] a problem that has later been addressed by the development of the fluorescent RNA aptamers Spinach[6], Broccoli[7], and Mango[8] that bind to non-toxic fluorophores. Spinach and Broccoli bind to the fluorophore 3,5-difluoro-4-hydroxybenzylidene imidazolinone (DFHBI), which mimics the fluorescence of GFP, whereas Mango binds the higher-wavelength fluorophore thiazole orange (TO) and derivatives hereof. Förster resonance energy transfer (FRET) is used experimentally to report on the conformation of biomolecules as it is especially sensitive towards changes in distance and orientation between a donor and acceptor fluorophore. FRET between FPs have been used to examine protein-protein interactions, where one of the first examples was a demonstration of the dimerization of the transcription factor Pit-1 by making fusion proteins of blue or green FPs with Pit-1[9]. In another study, a FP-based FRET system was used for creating a $Ca^{2+}$-sensor by linking the FPs with a $Ca^{2+}$-binding domain that changed conformation upon ligand-binding, thereby changing FRET[10]. Using similar strategies, other FP-based FRET sensors have been developed for both ions and small molecules[11], and these all share the advantages of being genetically encodable and providing ratiometric readouts that are not affected by fluctuations in sensor concentration.

FRET has not been used for RNA aptamer-based sensors, but in 2012, the Spinach aptamer was developed into an intensiometric riboswitch-type sensor by coupling the aptamer to an aptamer against a small molecule, namely S-adenosylmethionine (SAM)[12]. In this sensor, the presence of SAM was directly coupled to the detection of fluorescence from DFHBI bound in Spinach, and the sensor was shown to work both in vitro and in vivo. Similar sensors based on riboswitches have also been produced for cyclic di-GMP[13], cyclic di-AMP[14], and thiamine 5'-pyrophosphate[15]. The folding of such riboswitches has been studied using FRET, where regions of interest on the RNA were chemically labeled with small-molecule fluorophores making it possible to follow changes in conformation due to ligand-binding[16–18], however such labeled riboswitches are not expressible inside cells. In 2014, a reporter system for imaging gene expression in live cells, called IMAGEtag, was published. In this method, gene expression under different promotors were quantified by expressing two different RNA aptamers that bound externally added Cy3- or Cy5-modified ligands, providing an expressible system for real-time assessment of promotor activity[19]. Although this system was genetically encodable, it lacked structural control of the transcribed aptamers needed to make a system that could change conformation upon binding of a molecule of interest.

The research field of RNA nanotechnology aims at rationally designing RNA molecules that fold into well-defined shapes[20], and provides a tool for creating scaffolds that can spatially organize attached molecular components with high precision for application in, e.g., medicine[21, 22] and synthetic biology[23]. For example, multi-stranded RNA nanostructures have been used as isothermally self-assembled arrays to scaffold two proteins of the hydrogen biosynthesis pathway in E. coli and showed a 48-fold increase in hydrogen production[23]. Recently, the single-stranded RNA origami method, that allows rationally designing RNA

nanostructures to fold cotranscriptionally as they are being produced by the RNA polymerase, was introduced[24, 25]. The method was demonstrated in vitro, but compared to many other thermodynamically designed structures should be compatible with folding in the cell.

Here we demonstrate FRET between fluorescent RNA aptamers by positioning Spinach and Mango in close proximity on single-stranded RNA origami scaffolds. We use the aptamer-based FRET (apta-FRET) system for producing a dynamic and reversible RNA nanodevice and for making a SAM-sensor by also incorporating the SAM riboswitch in the structure. Finally, we demonstrate that the apta-FRET constructs are genetically encodable by observing FRET when expressing the constructs in E. coli.

## Results

**Design of the aptamer-based FRET system**. We set out to construct a FRET system based on fluorescent RNA aptamers and therefore examined the spectral properties of fluorophores bound by Spinach[6] (Fig. 1a) and Mango[8, 26] (Fig. 1b). DFHBI bound in Spinach and thiazole orange 3 (TO3-biotin) bound in Mango were found to have a significant spectral overlap of their emission and excitation spectra, respectively (Supplementary Fig. 1). The modified fluorophore DFHBI-1T displays brighter fluorescence compared to DFHBI and is slightly red-shifted to better fit with standard GFP filters[27]. In a similar effort, we synthesized an oxazole yellow (YO)[28] derivative, YO3-biotin (Fig. 1c) and found that both its excitation and emission spectra were ~40 nm blue-shifted as compared to TO3-biotin and have spectral properties very similar to those of Texas Red[29]. DFHBI-1T and YO3-biotin have an improved spectral overlap that is 32% larger compared to DFHBI and TO3-biotin (Fig. 1d; Supplementary Fig. 1), and their compatibility with standard filters makes this FRET pair widely applicable in fluorescence microscopy setups.

The Spinach and Mango aptamers were positioned on separate single-stranded RNA origami 2H-AE constructs[24] (2-helix constructs with anti-parallel even crossovers, Supplementary Note 1). In the Spinach construct, DFHBI-1T shows bright fluorescence even at low fluorophore and RNA concentrations (Supplementary Fig. 2a), whereas both the Mango construct and a 2H-AE construct without aptamers are unable to activate DFHBI-1T (Supplementary Fig. 2b, c). Thus, DFHBI-1T selectively binds only in the Spinach aptamer. Contrarily, YO3-biotin fluoresces with comparable intensities when bound in either the Mango or the Spinach construct (Supplementary Fig. 2d, e), similarly to what has been observed previously for TO1-biotin[30]. We further examined the binding affinities ($EC_{50}$) of the fluorophores in the two aptamers (Supplementary Table 1), and found that DFHBI-1T has a significantly lower $EC_{50}$ in Spinach than YO3-biotin (~340 nM compared to ~4 μM). Furthermore, we performed a competition assay demonstrating that DFHBI-1T outcompetes YO3-biotin bound in Spinach, whereas YO3-biotin is unable to outcompete DFHBI-1T bound in Spinach (Supplementary Fig. 3). Altogether, these findings indicate that Spinach preferentially binds DFHBI-1T when both fluorophores are present, and as Mango only binds YO3-biotin and with high affinity (26 nM), it suggests that the aptamers and fluorophores can be used as a FRET pair.

To create an aptamer-based FRET (apta-FRET) construct the Spinach and Mango aptamers were placed in close proximity in a single-stranded RNA origami 2H-AE construct (Fig. 1e). The apta-FRET constructs are annotated as SX-MY, where X and Y refers to the distance in base pairs from the crossover to Spinach (S) and Mango (M), respectively (Supplementary Note 1). When the fluorophores in Spinach and Mango were placed ~18 nm

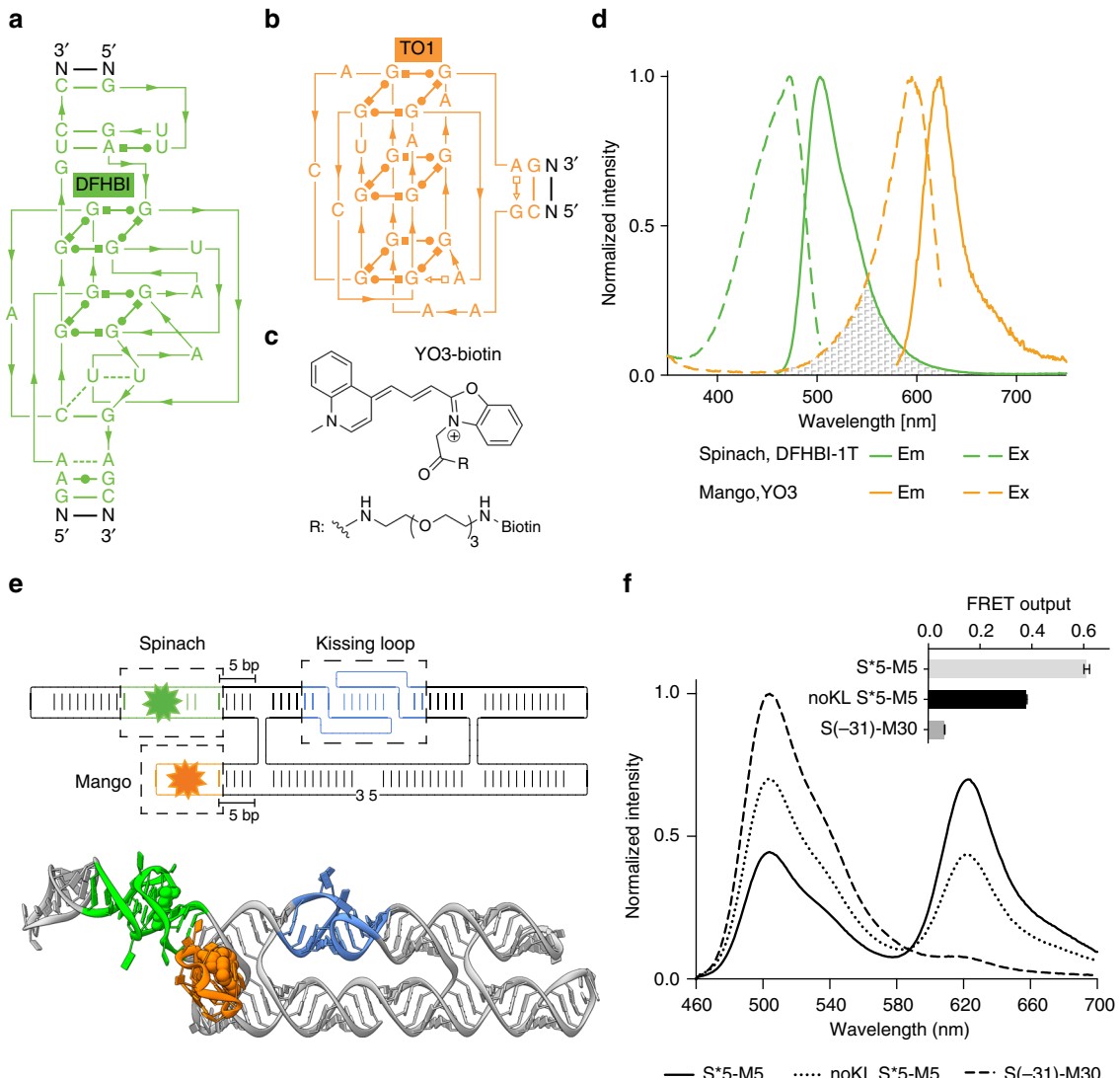

**Fig. 1** FRET between fluorescent aptamers on RNA origami scaffolds. **a** Structure diagram of the minimal core of the Spinach aptamer[31, 32] (green). The binding site of DFHBI-1T is indicated. Lines with arrows denote chain connectivity and Leontis-Westhof symbols[52] denote noncanonical base pairs. **b** Structure diagram of the minimal core of the Mango aptamer[26] (orange). The binding site of TO1-biotin is indicated[26]. **c** Chemical structure of YO3-biotin. **d** Excitation and emission spectra of DFHBI-1T and YO3-biotin in the Spinach and Mango construct, respectively. Spectral overlap between the emission spectrum of DFHBI-1T and the excitation spectrum of YO3-biotin is marked in gray. **e** 2D diagram and 3D model of an aptamer-based FRET (apta-FRET) construct showing the kissing-loop (blue), Spinach (green), and Mango (orange). The stem length between the construct crossover and both Spinach (S) and Mango (M) is 5 bps (and is thus named "S*5-M5", see text and Fig. 2 for further descriptions of nomenclature). The 3D model is shown in cartoon representation with fluorophores shown in space-fill representation. **f** Fluorescence spectra of apta-FRET constructs after excitation of DFHBI-1T. The solid line is the S*5-M5 construct, the dotted line is the S*5-M5 construct designed without a kissing-loop (KL), and the dashed line is the S(-31)-M30 construct designed to have no FRET. The inset shows calculated FRET outputs of the three constructs. Error bars indicate standard deviations calculated using triplicate measurements

from each other in the apta-FRET construct (S(-31)-M30), almost no fluorescence from YO3-biotin after excitation at DFHBI-1T wavelength was observed (Fig. 1f, dashed line), and the construct therefore had very low FRET output (Fig. 1f, insert). However, when both aptamers were incorporated in close proximity in the apta-FRET construct (S*5-M5), YO3-biotin displayed bright fluorescence after excitation at DFHBI-1T wavelength (Fig. 1f, solid line), indicating that this construct has high FRET (Fig. 1f, insert). We further designed a construct without the internal kissing-loop (KL) (shown in blue in Fig. 1e), which otherwise stabilizes the structure. This caused a ~40% decrease in FRET (Fig. 1f, dotted line and insert), which indicates that the aptamers are further apart, demonstrating a conformational difference between the apta-FRET constructs with and without the

stabilizing KL. Thus, the apta-FRET system should be very sensitive to conformational changes in the RNA origami scaffold.

As we set out to develop a genetically encodable FRET system where the apta-FRET constructs could be expressed in vivo, we also investigated the in vivo optimized version of the Spinach aptamer, called Broccoli[7]. However, when testing the aptamers inserted in our RNA origami structures in vitro, Spinach performed as well as Broccoli for fluorescence enhancement (Supplementary Fig. 4) and yielded higher FRET outputs than structures containing Broccoli (Supplementary Fig. 5).

**Optimization of the FRET efficiency.** In the presented apta-FRET constructs, the fluorophore DFHBI-1T has a distinct

orientation as the Spinach aptamer is rigidly folded and the fluorophore is specifically bound[31, 32]. The Mango aptamer also binds its fluorophore in an orientation-specific manner, and the aptamer has a specific fold, but a recent publication of the co-crystal structure of Mango[26] revealed a flexible link between the fluorophore-binding G-quadruplex and the tetraloop-like motif that connects it to the double helix domain. Therefore, the Mango aptamer and its fluorophore have some freedom of rotation compared to the more rigid Spinach aptamer and its fluorophore. This is important when performing FRET measurements, as the relative orientation of the dipole moments of the two fluorophores influence the efficiency of energy transfer. To increase the chance of obtaining high FRET we calculated the dipole moment of DFHBI-1T bound in Spinach and used 3D modeling (see Methods section) to point the dipole moment either towards or away from the helix axis of Mango (Fig. 2), which, according to dipole moment theory, causes efficient energy transfer[33]. Unfortunately, Trachman et al.[26] were not able to model TO3-biotin into the binding pocket in the co-crystal structure of Mango without steric clashes. Therefore, the precise position of YO3-biotin in Mango is unknown and the direction of its dipole moment could not be used in the 3D modeling. To obtain a construct with a high FRET value we investigated two different orientations of Spinach and different lengths of the Mango stem (Fig. 2). By fixating Spinach in the position where the dipole moment pointed towards the helix axis of Mango (Fig. 2a), and by varying the spacer stem length of Mango, we found maximal

FRET values when the spacer stem length was either 16–20 or 30 bps (Fig. 2b, solid line). When measuring the distance between the fluorophores in the 3D model (Fig. 2b, dashed line), it was found that the highest FRET values corresponded to the shortest distances, as is expected in FRET. In some cases, the 3D model displayed steric clashes when positioning Mango with those specific stem lengths and in these cases the distances are underestimated (Fig. 2b, red circles). However, the apta-FRET constructs seem to fold correctly despite the steric clashes, which could be due to the aforementioned flexibility of the Mango aptamer. When Spinach was flipped[34] (annotated as S*, e.g., S*16-M18), such that the dipole moment of DFHBI-1T pointed away from the helix axis of Mango (Fig. 2c), and the length of the spacer stem of Mango was varied, maximal FRET values were observed for a spacer stem length of 16–18 bps (Fig. 2d). Again, we found a similar correlation between FRET outputs and the distances between the two fluorophores. The two constructs with the highest FRET outputs (S17-M30 and S*16-M16) were used as guides to create minimal versions, which had a complete helix turn removed from each spacer region (S6-M19 and S*5-M5). These minimal versions showed even higher FRET (Supplementary Fig. 5), which indicate some flexibility in the longer stems. In conclusion, we have optimized the relative orientation and position of the fluorophores and found constructs with high FRET values, which demonstrates that single-stranded RNA origami can be used to position RNA aptamers with high precision.

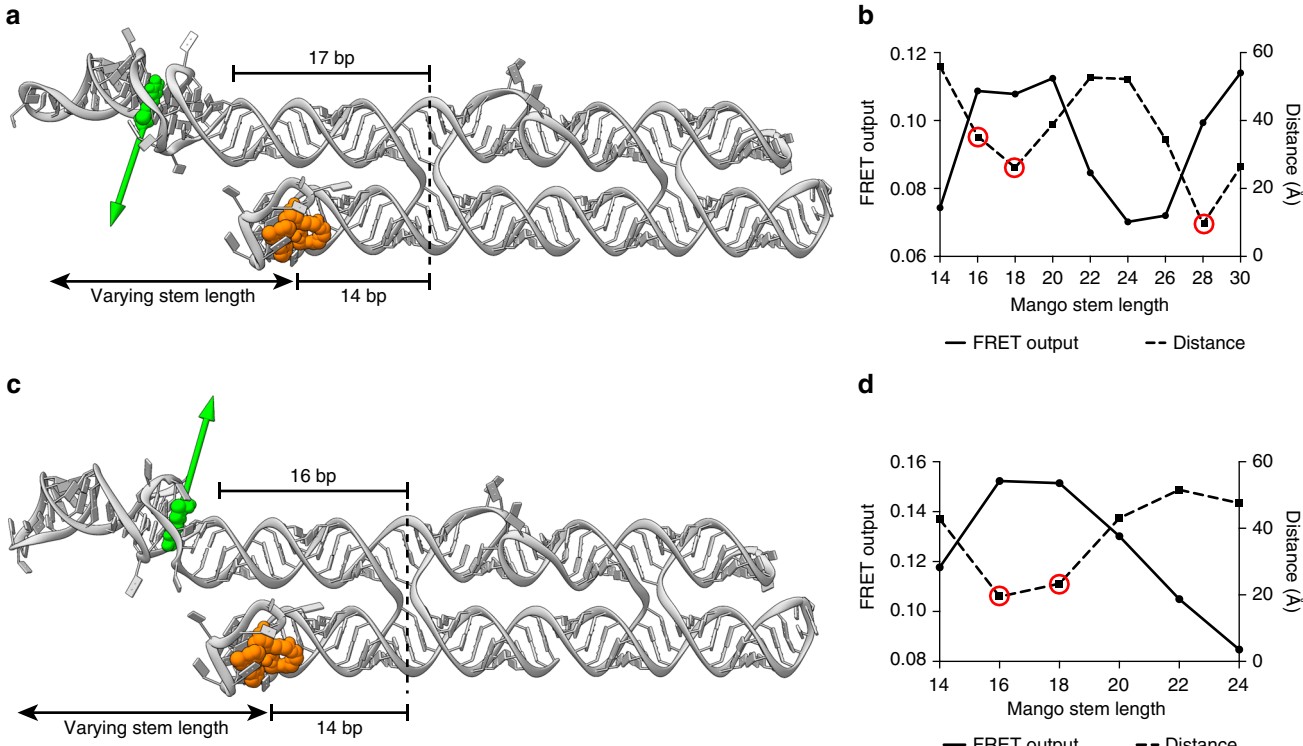

**Fig. 2** Influence on FRET of dipole moment orientation and distance between Spinach and Mango. **a** 3D structure of the apta-FRET construct in which Spinach is placed in the construct in its normal orientation. DFHBI-1T (green) and TO1-biotin (orange) is shown in space-fill representation, and the calculated dipole moment of DFHBI-1T is shown as a green arrow. As the specific binding of YO3-biotin in Mango is unknown, no dipole moment is indicated. **b** FRET output of the S17-M14 to S17-M30 constructs (the number after S and M denotes the distance in base pairs between the crossover and Spinach and Mango, respectively). The distances between DFHBI-1T in Spinach and TO1-biotin in Mango measured on 3D models are also shown. The red circles around some distance points denote 3D models in which the Mango aptamer sterically clashes into the stem carrying Spinach. **c** 3D structure of the apta-FRET construct in which Spinach is placed in the construct in a flipped (*) orientation. **d** FRET output and distances between DFHBI1-T and TO1-biotin of the S*16-M14 to S*16-M24 constructs. Red circles similar to **b**

**Observing conformational changes using the apta-FRET system**. To investigate the ability of the apta-FRET system to respond to conformational changes, we constructed an RNA nanomechanical device that, upon binding of an invading RNA oligonucleotide, changes its conformation. In this apta-FRET construct, the internal KL was extended at the A-bulge with a hairpin to create a "branched kissing-loop" motif (Fig. 3a), which was inspired by the Z-motif developed by Di Liu and Yossi Weissman[35]. The hairpin was made complementary to an invader RNA strand, where the hairpin loop functions as a toehold for initial invader binding that leads to disruption of the stem as well as of the internal KL interaction by strand displacement[36]. On the basis of the S*5-M5 construct, we designed two apta-FRET constructs that respond to different RNA invaders (Supplementary Note 1). The apta-FRET system was able to change conformation as a response to both RNA invaders as observed by a decrease in FRET output upon addition of invader RNAs (Fig. 3b). Furthermore, the apta-FRET system showed no

response when an unrelated RNA sequence was added (Supplementary Fig. 6). To demonstrate reversibility, the invader RNAs contained a toehold that subsequently could be used to remove the invader RNAs using complementary "anti-invaders", thus allowing the KL to reform and restore an increased FRET signal. Time course measurements (Fig. 3b) show that the apta-FRET system restores more than 80% of the initial FRET signal upon addition of anti-invaders. The experiment demonstrates that the apta-FRET system can be used to report on conformational changes in the structure.

To demonstrate the versatility of the apta-FRET system, we created a FRET-based conformational sensor that responds to the small-molecule $S$-adenosylmethionine (SAM). A SAM riboswitch[37] was integrated into the S6-M19 construct to create the S6-M19-SAM sensor (Supplementary Note 1, see the Methods section for design details). In the absence of SAM the riboswitch is expected to be in a flexible and partially unstructured conformation[38, 39] (Fig. 3c, top), and upon binding of SAM, it

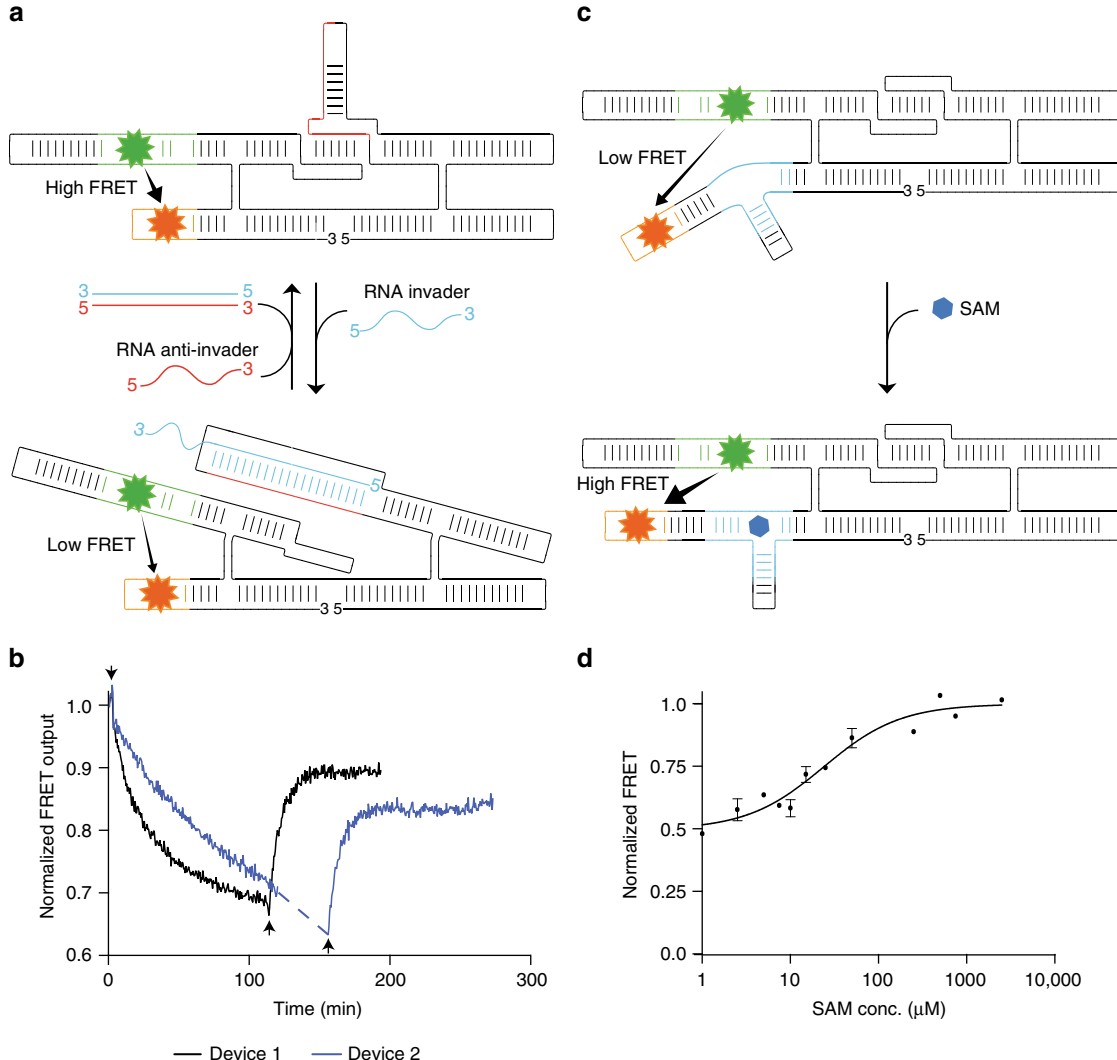

**Fig. 3** Studying conformational changes using the apta-FRET system. **a** Secondary structure representation of an apta-FRET conformational switch containing Spinach (green), Mango (orange), and the KL extended with a hairpin, which has a seed sequence (red) that allows strand invasion of the KL by specific RNA sequences (blue). **b** Time-resolved FRET measurements of the reversible invasion of the apta-FRET conformational switch with toehold extended invader RNAs and their anti-invaders. The arrow pointing down denotes the addition of toehold extended invader RNA and the arrows pointing up denote addition of anti-invaders. **c** The partially unstructured SAM riboswitch (light blue) in the apta-FRET SAM sensor causes a low FRET signal, when SAM (blue) is not present. Upon binding of SAM, the riboswitch motif transforms to a more structured conformation and the average distance between the fluorophores in Spinach (green) and Mango (orange) is decreased, which results in an increased FRET signal. **d** Dose-response curve of the apta-FRET SAM sensor with an EC$_{50}$ of 17 µM. Error bars indicate standard deviations ($n = 3$)

changes to a more rigidly structured conformation (Fig. 3c, bottom). This conformational change was observed upon SAM binding as an increase in the FRET signal of the apta-FRET SAM sensor (Fig. 3d). The EC$_{50}$ of the sensor was calculated from the dose–response curve to be 17 μM. To verify that changes in FRET signal was caused by the intended conformational change and not by an unintentional perturbation of Mango's binding affinity towards YO3-biotin caused by the integration of the riboswitch, the fluorescence from YO3-biotin under direct excitation was also recorded (Supplementary Fig. 7). We observed no change in the fluorescence of YO3-biotin over the range of SAM concentrations tested. The experiment demonstrates that the apta-FRET system can be used to report on conformational changes in a riboswitch incorporated in the RNA origami structure, thereby producing a FRET sensor.

**Expressing the apta-FRET system in *E. coli* cells.** One of the greatest advantages of single-stranded RNA origami is that rationally designed structures can fold cotranscriptionally, and thus should be able to form in vivo. To demonstrate that the

apta-FRET constructs are genetically encodable, we transformed *E. coli* cells with plasmids coding for one of four different constructs; S*5-M5, B*5-M5, S(-31)-M30 and an unmodified 2H-AE structures (Supplementary Note 1). The 2H-AE structure was used as a negative control and it showed significantly lower fluorescence compared to the apta-FRET constructs. Both S*5-M5 and B*5-M5 were tested to compare FRET outputs of these constructs as Broccoli is optimized for in vivo work, but constructs containing Spinach gave higher FRET outputs in vitro. The four constructs were analyzed using flow cytometry and spectrofluorometric measurements (Fig. 4a; Supplementary Figs. 8 and 9). *E. coli* cells analyzed using both methods revealed that the S*5-M5 construct had the highest FRET signal, B*5-M5 slightly lower, and S(-31)-M30 displayed almost no FRET. These results are comparable to the in vitro results using the same constructs (Fig. 1f). *E. coli* cells expressing the S*5-M5 or S(-31)-M30 constructs were visualized in a confocal microscope and cells that had taken up both fluorophores were selected for single cell analysis (Fig. 4b, c; Supplementary Fig. 10). The S*5-M5 construct displayed the highest FRET outputs with values centered around 0.19, whereas the S(-31)-M30 had lower FRET outputs with the

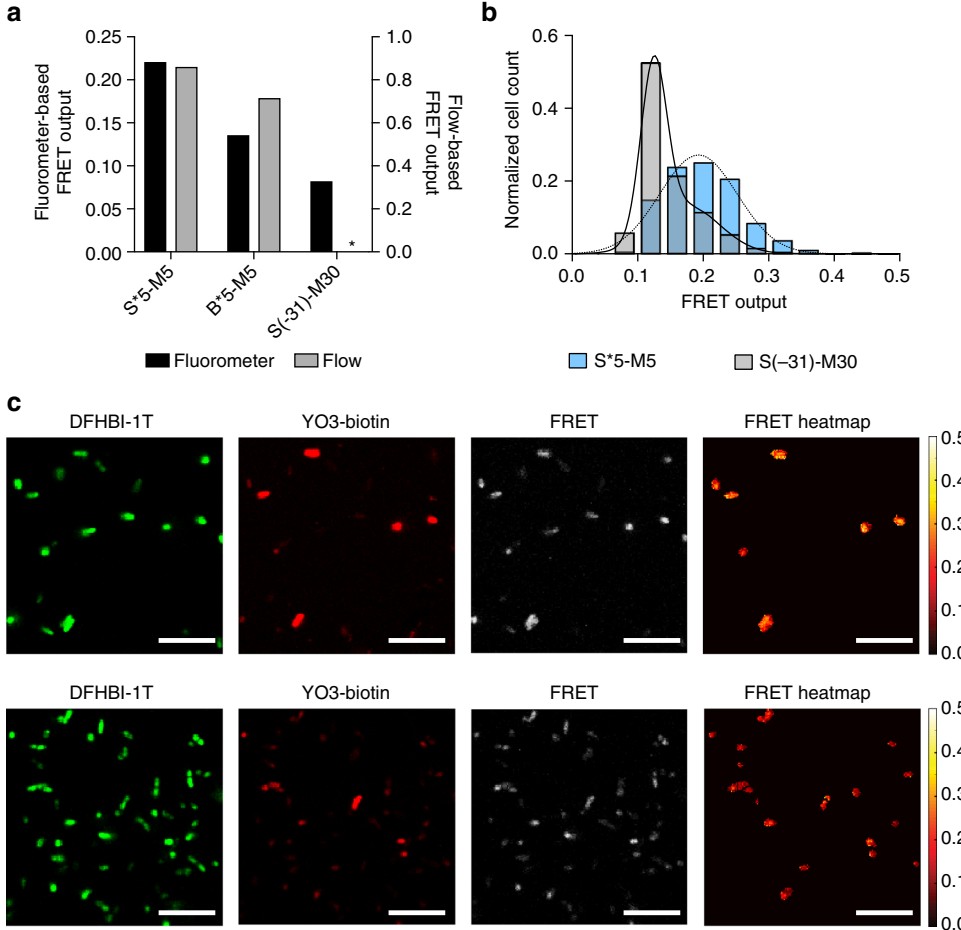

**Fig. 4** Studies of apta-FRET constructs expressed in *E. coli*. **a** FRET outputs of S*5-M5, B*5-M5, S(-31)-M30, and 2H-AE constructs. Black bars are FRET outputs calculated based on spectrofluorometric measurements, gray bars are calculated based on flow cytometry measurements. The asterisk denotes a FRET output that is effectively zero (see Supplementary Table 3). **b** Single cell analysis of confocal microscopy images of the S*5-M5 (blue) and S(-31)-M30 (gray) constructs. The analysis is based on 10 images of each construct. The solid line is a sum of two Gaussians fitted to the S(-31)-M30 FRET outputs; the dotted line is a single Gaussian fitted to the S*5-M5 FRET outputs. **c** Confocal microscopy and FRET heatmap images of cells expressing the S*5-M5 (top) and S(-31)-M30 (bottom). In the FRET heatmap images to the right, red colors indicate low FRET outputs and yellow colors indicate high FRET outputs. Only bacteria displaying fluorescence from both DFHBI-1T and YO3-biotin were selected for representation in the heatmap (see Methods section and Supplementary Fig. 10 for further details on the selection process). The scale bars correspond to 10 μm

largest population of cells centered around 0.12. A FRET heatmap image (Fig. 4c, right) based on the selected cells visually confirmed that cells expressing S*5-M5 had higher FRET output. Taken together, these results confirm the correct folding of single-stranded RNA origami structures in vivo, and show that the apta-FRET system functions properly in *E. coli*.

## Discussion

In summary, we have developed a FRET system based on fluorescent RNA aptamers that is genetically encodable and can report on conformational changes. We used single-stranded RNA origami to position two fluorescent RNA aptamers in close proximity resulting in high FRET between the two fluorophores. The development of a new fluorophore allowed us to further increase the FRET output. By incorporating RNA motifs that responded to either small RNA strands or a small molecule, we demonstrated the apta-FRET systems ability to report on conformational changes. Finally, we successfully expressed the apta-FRET system in *E. coli* cells, which confirmed that the system is genetically encodable and functioning in vivo.

Rationally designed RNA nanostructures and devices that can be genetically encoded and function inside cells have great potential for applications in synthetic biology, metabolic engineering, and nanomedicine[40, 41]. To be expressed in cells, designed RNA nanostructures have to be folded during transcription similarly to natural RNA structures, and co-transcriptional folding of RNA nanostructures has been demonstrated in vitro[24, 25]. Although RNA nanostructures have been used as expressible scaffolds for organizing proteins in *E. coli* cells[23] the direct verification of correctly folded RNA nanostructures in vivo remains difficult due to the lack of experimental techniques to study RNA folding inside living cells. We anticipate that the developed apta-FRET system can be used as a general tool for in vivo studies of the folding and conformational changes of both natural and artificial RNA structures. Folding studies based on apta-FRET can be used to verify the correct folding of more complex rationally designed RNA nanostructures in cells, or to study the global conformation of naturally occurring RNA structures. As demonstrated in vitro, in this study, conformational changes can be observed for dynamic RNA nanodevices and may in the future allow detection of functional RNA devices in vivo. The apta-FRET system may also be grafted onto natural RNA machines like the ribosome to observe conformational changes in the cell. Inspired by FRET-based conformational sensors using fluorescent proteins, similar RNA-based FRET conformational sensor devices may be developed. Thus, we see the apta-FRET system as an important tool for developing genetically encodable RNA scaffolds, sensors, and devices for advanced applications in synthetic biology.

## Methods

**YO3-acetate synthesis**. Et$_3$N (0.26 mL, 1.86 mmol, 1 eq) was added to a stirred solution of 4-[(E)-2-(acetylphenylamino)ethenyl]-1-methylquinolinium iodide[28] (790 mg, 1.84 mmol, 1 eq) and 2-(2-methylbenz[d]oxazole-3-ium-3-yl) acetate[42] (500 mg, 2.62 mmol, 1.42 eq) in DCM. The solution was left to stir for 20 h followed by in vacuo concentration. The solid was then re-suspended in DCM (150 mL), filtered and washed with acetone (3 × 10 mL). The solid was dried in vacuo to yield (85 mg, 12%) YO3-acetate as an amorphous purple solid. $^1$H NMR (400 MHz, DMSO-$D6$) δ 10.17 (s, 1H), 8.47–8.38 (m, 2H), 8.34–8.28 (m, 1H), 8.24–8.18 (m, 1H), 8.18–8.13 (m, 1H), 7.94–7.87 (m, 2H), 7.68–7.59 (m, 3H), 7.52–7.46 (m, 1H), 7.41–7.28 (m, 3H), 7.01–6.94 (m, 1H), 5.91–5.82 (m, 1H), 4.47 (s, 2H), 4.05 (s, 3H). HRMS (ESI) [M$^+$,C$_{22}$H$_{19}$N$_2$O$_3^+$] requires $m/z$ 359.1390, found 359.1395.

**YO3 PEG 3 link amine biotin synthesis**. HBTU (45 mg, 3 eq) and DIPEA (30 eq) was added to YO3-acetate (3 mg, 1 eq) in DMF (250 µL). The solution was allowed to stir for 10 min where after N-biotinyl-3,6,9-trioxaundecane-1,11-diamine[43] (7 mg, 2 eq) was added and the solution was stirred for 18 h. The solvent was lyophilized and the residue was redissolved in 1:1 MeCN/50 mM NH$_4$OAc$_{(aq)}$. The

solution was then purified by RP-HPLC (25–50% MeCN against 50 mM NH$_4$OAc over 30 min). RP-HPLC Rt = 14.9 min. HRMS (ESI) [M$^+$,C$_{40}$H$_{51}$N$_6$O$_7$S$^+$] requires $m/z$ 759.3534, found 759.3531.

**Structural RNA modeling**. The three-dimensional (3D) design of the basic 2H-AE RNA origami structure has previously been described in great detail[24, 44]. Briefly, using the 3D modeling programs Swiss-PdbViewer[45] and UCSF Chimera[46] two standard A-form RNA double helices were aligned in parallel and rotated to create optimal spacing for an anti-parallel even (AE) double crossover (DX). An internal 180° kissing-loop (HIV-1 DIS, PDB id: 2B8R) was positioned between the crossovers and UUCG tetraloops (extracted from PDB id: 1F7Y) were positioned at the ends of the double helices. The 3D design of the apta-FRET constructs was based on the 2H-AE RNA origami structure. The minimal core of the Spinach aptamer (PDB id: 4Q9R), previously used to create Bunch of Baby Spinach[47], was positioned on one of the outer double helix domains flanking the DXs. The Spinach motif was rotated to point the dipole moment of DFHBI-1T either directly towards or away from the adjacent parallel-aligned double helix domain, where the Mango aptamer was to be positioned. The core in the original Spinach aptamer is flanked by two double helix domains called P1.2 and P2.1 by Huang et al.[51] This allows two different versions of incorporation; the "normal" version where the P1.2 domain was used to connect Spinach to the RNA origami, and the "flipped" version (annotated with *) where the P2.1 domain was used to connect Spinach to the RNA origami. A stem-loop built from a standard A-form double helix and a UUCG tetraloop was attached to the other end of the Spinach core. Due to steric clashes only two of the possible four versions were viable; the normal version with the dipole pointing towards the adjacent double helix, and the flipped version with the dipole pointing away from the adjacent double helix. The recently published co-crystal structure of the Mango aptamer revealed a three tier G-quadruplex connected to a double helix domain through a GAAA tetraloop-like motif, and does not allow Mango to be "flipped" like Spinach. The linkage between the tetraloop and G-quadruplex is not structurally defined and is expected to be flexible, and TO3-biotin (very similar to YO3-biotin) could not be modeled into the binding pocket without steric clashes. Therefore, we could not model the precise position of the dipole of the YO3-biotin fluorophore. To find the optimal position of Mango the length of the double helix domain between Mango and the nearest DX was varied in steps of two base pairs. Some of the positions were found to have steric clashes. After initial modeling 3D structures were ligated with a Perl script ("ligate. pl", which is available from our webpage, www.andersen-lab.dk) and refined using a recursive geometric refinement function in Assemble2[48]. 3D models in this work were rendered in UCSF Chimera.

**Dipole moment calculation**. The dipole moment of DFHBI-1T was calculated using the Marvin software suite (version 15.10.19) and the Calculator Plugin developed by ChemAxon (http://www.chemaxon.com/). The chemical structure of DFHBI-1T was drawn in MarvinSketch, and the protonation state of the molecule was determined at pH 7.8 by using the pKa Calculator Plugin. The dipole moment of the most abundant DFHBI-1T microspecies (70.4% at pH 7.8) was calculated by using the Dipole Moment Calculator Plugin. The total dipole moment of DFHBI-1T was visualized as a vector expressed in the principal axis frame. The resulting file indicating both the chemical structure and dipole moment vector of DFHBI-1T was exported as pdb file and imported in Swiss-PdbViewer[45] to align the fluorophore molecule with the 3D coordinates of DFHBI from the crystal structure of Spinach (PDB id: 4Q9R).

**RNA sequence design**. The sequence design process has previously been described in detail[24, 44]. Briefly, 3D models were loaded in Assemble2[48] to visualize the secondary structure, which were transferred to text-based 2D structure design files by hand. For the apta-FRET with a branched KL a T-junction motif was created in the 2D blueprint by exchanging the two bulged-out adenines in one of the kissing-loop partners with a stem-loop. The sequence of the conserved RNA motifs (KL, tetraloops, Spinach, Mango, RNA invader) was retained and the remaining sequence was denoted with Ns. Each sequence was further constricted by "locking" the crossovers with G-C base pairs, incorporating a GU wobble pair per every eight continuous base pairs, and constraining the 5′-end to 5′-GGGAGA-3′, an optimal initiation sequence for T7 RNA polymerase. The 2D blueprints were read by a Perl script ("trace.pl", which is available from our webpage, www.andersen-lab.dk) that outputs the dot-bracket notation and sequence constraints appropriate for sequence design in NUPACK[49]. Sequences designed in NUPACK were folded and evaluated using mFold[50].

**Synthesis of DNA templates**. DNA templates of the apta-FRET constructs were produced by standard PCR amplification of gBlocks gene fragments from IDT using Phusion High-Fidelity DNA Polymerase (ThermoFischer Scientific). Amplifications were performed in 1× Phusion HF buffer (provided by the manufacturer), 200 µM dNTP (invitrogen), 1 µM forward (Fwd) and reverse (Rev) primers (ordered from IDT), 4 ng gBlock and 1 U Phusion polymerase. DNA templates of the simple aptamers were produced by standard PCR amplification of DNA ultramers (IDT) using *Taq* DNA polymerase (Invitrogen). Amplifications were performed in 1× *Taq* buffer, 200 µM dNTP (invitrogen), 1.5 mM MgCl$_2$, 1 µM

Fwd and Rev primers, 4 ng ultramer, and 2.5 U *Taq* DNA polymerase. Fwd and Rev primers were designed for each template (see Supplementary Note 2), and annealing temperatures were calculated using New England Biolabs (NEBs) $T_m$ calculator (http://tmcalculator.neb.com/). The amplified DNA was purified using GFX PCR DNA purification kit (illustra) and stored in TE buffer.

**In vitro transcription and purification of RNA**. In vitro transcription was performed by mixing DNA template, transcription buffer (15 mM Mg(OAc)$_2$, 50 mM Tris-Acetate pH 7.8, 40 mM NaCl, 0.1% Tween20), 1 mM DTT, and NTPs (2.5 mM each). T7 RNA polymerase (made in-house) was added, and the samples were incubated at 37 °C for at least 4 h. The reaction was stopped by adding DNase I (ThermoFischer Scientific) to a concentration of 1 U/100 μL and incubating at 37 °C for 15 min. The transcribed RNA was purified on 6% polyacrylamide gels after mixing 1:1 with denaturing load buffer (95% formamide, 18 mM EDTA, 0.025% SDS, bromophenol blue, xylene cyanol) and heating to 95 °C for 5 min. Gel bands were visualized using UV shadowing, cut out of the gel, and eluted overnight in nuclease free H$_2$O (Ambion). Gel pieces were separated from the liquid using Freeze'n'Squeeze DNA gel extraction Spin columns (Bio-Rad), and the RNA was finally ethanol precipitated, re-suspended in nuclease free H$_2$O and stored at −20 °C.

**Heat annealing of RNA origami structures**. The transcribed RNA was heat annealed prior to usage to ensure correct folding of the RNA origami structures. First, the RNA in nuclease free H$_2$O was heated to 95 °C for 5 min, and then quickly cooled at −20 °C for 3 min. 5× assembly buffer (5X TB, 625 mM KCl, 5 mM MgCl$_2$) was then added to a final concentration of 1×, and the samples were left at 37 °C for 30 min.

**Fluorescence measurements**. Fluorescence measurements were performed on a FluoroMax 4 from Horiba, Jobin Yvon. Excitation of DFHBI-1T and YO3-biotin were performed at 450 nm and 580 nm, respectively. Emission of DFHBI-1T and YO3-biotin were recorded at 503 nm and 620 nm, respectively. Monochromator slits were set to 5 nm, and integration time was 0.2 s. Measurements were performed on sample volumes between 65 and 100 μL, and fluorophore concentrations of 1 μM DFHBI-1T and 1.7 μM YO3-biotin were used if nothing else is stated. Relative FRET values were calculated using the equation $FRET = \frac{I_{DY}(ex_D, em_Y)}{I_{DY}(ex_D, em_Y) + I_{DY}(ex_D, em_D)}$, where $I_{DY}(ex_D, em_Y)$ is the emission at YO3-biotin wavelength after DFHBI-1T excitation and $I_{DY}(ex_D, em_D)$ is the emission at DFHBI-1T wavelength after DFHBI-1T excitation. Both were measured with DFHBI-1T and YO3-biotin present.

**Fluorophore binding and competition assays**. Fluorophore activation and competition assays were performed on RNA origami structures containing only Spinach or only Mango, or with an empty 2H-AE structure (Supplementary Note 1). The constructs were heat annealed prior to measurements according to the protocol above. Concentrations of both RNA and fluorophores are stated in the relevant figures. Measurements were performed in 1× assembly buffer.

**Fluorophore EC$_{50}$ measurements**. The binding affinities towards different fluorophores of the aptamers integrated into RNA origami structures were measured. The constructs were heat annealed prior to use according to the protocol above, and samples prepared in 1× assembly buffer. Measurements were performed using serial diluted fluorophores of concentrations between 1 nM and 2 μM. All samples were prepared in triplicates both with and without 20 nM RNA, and fluorescence was measured on a VarioSkan Flash Reader (Thermo Scientific) by exciting the samples containing DFHBI-1T or YO3-biotin at 450 nm or 580 nm, respectively. Background fluorescence from each concentration of fluorophore in the absence of RNA was subtracted from the recorded intensities before calculating EC$_{50}$. EC$_{50}$ was calculated in GraphPad Prism 7.0b for Mac OS X, using the non-linear regression analysis "log (agonist) vs. response (three parameters)" of a dose–response curve.

**Fluorescence enhancement measurements**. Enhancement of fluorescence was calculated for the fluorophores in the minimal version of the aptamers and in the aptamers integrated into RNA origami structures. Overall, 1 μM RNA in 1× assembly buffer was incubated with 100 nM of the relevant fluorophore. Fluorescence enhancement was calculated using the equation $F_E = \frac{I_{F+R}}{I_F}$, where $I_{F+R}$ is the fluorescence intensity of the fluorophore in the presence of RNA and $I_F$ is the fluorescence intensity of the fluorophore alone (off-state fluorescence).

**Conformational changes in the apta-FRET system**. Visualization of conformational changes as a response to invader RNAs was performed by time-resolved measurements of the apta-FRET system containing a branched KL. Fluorescence both from DFHBI-1T and YO3-biotin was recorded every 30 s, and the shutters were closed between measurements to avoid bleaching. Reversible invasion was obtained by first adding invader RNA with a toehold to a final concentration of 200 nM to a mixture of 100 nM apta-FRET structure, 1 μM DFHBI-1T, and 1.7 μM YO3-biotin, and subsequently adding "anti-invader" to a final concentration of 400 nM. FRET outputs were calculated as described previously.

Invader 1: UAGCUUAUCAGACUGAUGUUGAUAUAAAAG;

Anti-invader 1: CUUUUAUAUCAACAUCAGUCUGAUAAGCUA.
Invader 2: CUAGACUGAAGCUCCUUGAGGGAAGUUAG;
Anti-invader 2: CUAACUUCCCUCAAGGAG CUUCAGUCUAG.

**EC$_{50}$ measurements of SAM in the apta-FRET SAM sensor**. The binding affinity of SAM in the S6-M19-SAM sensor was performed by incubation with serial diluted SAM of concentrations between 1 μM and 2.5 mM. The measurements were performed in triplicates with 50 nM RNA in assembly buffer containing 1 μM DFHBI-1T and 1.7 μM YO3-biotin. Measurements were performed in a VarioSkan Flash Reader by exciting the samples at 450 nm and recording fluorescence spectra from 470 nm to 700 nm. FRET outputs and EC$_{50}$ were calculated as described previously.

**Cloning procedure of apta-FRET constructs**. Standard molecular cloning steps were performed. First, amplification and encoding of restrictions sites (*Bgl*II and *Eco*RI) onto the ends of the DNA templates were performed by PCR amplification using Phusion DNA polymerase as described above (see Supplementary Note 2 for primer sequences). After digestion with *Bgl*II and *Eco*RI (ThermoFischer scientific —Fastdigest) for 30 min at 37 °C, the DNA was purified using GFX PCR purification kit (illustra). pET28c-F30–2xdBroccoli was received as a gift from Samie Jaffrey (Addgene plasmid # 66843). A plasmid prepared in-house based on the pET28c plasmid was digested with *Bgl*II and *Eco*RI, and purified on a 1% 0.5× TBE agarose gel. The linearized pET28c plasmid was extracted from gel using GFX gel extraction kit (illustra) and the insert was ligated into the plasmid in a 1:3 ratio by incubation with T4 DNA ligase at 16 °C overnight. The ligation reaction mixture was used directly to transform the plasmid into DH5-alfa *E. coli* (NEB) using standard heat-shock procedure. The transformed bacteria were plated onto LB-agar with Kanamycin (Kan, 50 μg/mL), and incubated overnight at 37 °C. Single colonies were selected and inoculated in LB-Kan (50 μg/mL) overnight at 37 °C with vigorous shaking. The plasmids were isolated using GeneJET plasmid miniprep kit (ThermoFischer Scientific) before verified by sequencing at GATC-Biotech. Sequence positive plasmids (~1 μg) were transformed into chemically competent HT115(DE3) *E. coli* made in-house, and the positive transform clones were selected using LB-agar plates with Kan (50 μg/mL) and Tetracyclin (Tet, 12.5 μg/mL) as described in Hull and Timmons[51].

**Culture procedure and setup**. An overnight (ON) culture was made by inoculating single HT115(DE3)-pET28c-FRET colonies in 3 mL LB with Kan (50 μg/mL) and Tet (12.5 μg/mL) and allowed to grow aerobically at 37 °C with vigorous shaking overnight. The culture was diluted 1:100 in LB-Kan and incubated using the same conditions until OD$_{600}$ ~0.25–35 (~3 h). Then IPTG (1 mM) was added to half of the sample, and the cultures incubated for ~6–9 h. DFHBI-1T (50 μM) and YO3-biotin (85 μM) were added and the cultures incubated for 30 min. The different cultures were pelleted by centrifugation at 4000×g for 5 min before re-suspended in fresh 1× M9 salt (Sigma#M6030) and analyzed by spectrofluorometry, flow cytometry or confocal microscopy.

**Spectrofluorometric measurements**. The cells prepared as described above were analyzed by spectrofluorometric measurements in a volume of 50 μL Samples containing both DFHBI-1T and YO3-biotin as well as controls containing only DFHBI-1T or YO3-biotin were prepared. Excitation of DFHBI-1T and YO3-biotin were performed at 450 and 580 nm, respectively. FRET outputs were calculated by first calculating the direct acceptor excitation, which was found to be 1.5% (see Supplementary Table 2) of the total YO3-biotin emission using the equation: $A_{dir} = \frac{I_Y(ex_D, em_Y)}{I_Y(ex_Y, em_Y)}$, where $I_Y(ex_D, em_Y)$ is the emission at YO3-biotin wavelength after DFHBI-1T excitation with only YO3-biotin present, and $I_Y(ex_Y, em_Y)$ is the emission at YO3-biotin wavelength after YO3-biotin excitation with only YO3-biotin present. Hereafter, the emission intensity arising from the DFHBI-1T tail at YO3-biotin wavelength was found; it was calculated to 7.4% (Supplementary Table 2) of the maximum DFHBI-1T emission using the equation: $D_{leak} = \frac{I_D(ex_D, em_Y)}{I_D(ex_D, em_D)}$, where $I_D(ex_D, em_D)$ is the emission at DFHBI-1T wavelength after DFHBI-1T excitation with only DFHBI-1T present, and $I_D(ex_D, em_Y)$ is the emission at YO3-biotin wavelength after DFHBI-1T excitation with only DFHBI-1T present. FRET outputs were hereafter calculated by subtracting both $A_{dir}$ and $D_{leak}$ using the equation: $FRET = \frac{I_{DY}(ex_D, em_Y) - 0.015 \cdot I_{DY}(ex_Y, em_Y) - 0.074 \cdot I_{DY}(ex_D, em_D)}{I_{DY}(ex_D, em_Y) - 0.015 \cdot I_{DY}(ex_Y, em_Y) - 0.074 \cdot I_{DY}(ex_D, em_D) + I_{DY}(ex_D, em_D)}$, where $I_{DY}(ex_D, em_Y)$ is the emission at YO3-biotin wavelength after DFHBI-1T excitation, $I_{DY}(ex_Y, em_Y)$ is the emission at YO3-biotin wavelength after YO3-biotin excitation, and $I_{DY}(ex_D, em_D)$ is the emission at DFHBI-1T wavelength after DFHBI-1T excitation, all three with both DFHBI-1T and YO3-biotin present.

**Flow cytometry**. The same samples as used for spectrofluorometric measurements (~50 μL) were diluted in 950 μL 1× M9 salt before measured on a Beckmann Coulter Gallios Flow Cytometer with a 488 nm laser and the FL1(575/30) and FL3(620/30) detector filters. Kaluza Analysis software was used to process the data. Fluorescence emission was measured on 50,000 ungated cells either containing both DFHBI-1T and YO3-biotin or containing only DFHBI-1T or YO3-biotin. The geometric mean intensities of gated 2H-AE cells were used as a background and subtracted from the geometric mean intensity measured on gated cells containing apta-FRET structures. The

intensity arising from DFHBI-1T in the FL3 channel was found using $D_{leak} = \frac{FL3_D}{FL1_D}$, where $FL3_D$ and $FL1_D$ are the geometric mean intensities in the FL3 and FL1 channels, respectively, from cells only containing DFHBI-1T. $D_{leak}$ was found to be 9.15 % on average. FRET outputs were calculated using the equation: $FRET = \frac{FL3_{DY} - FL3_Y - 0.09^* FL1_{DY}}{FL3_{DY} - FL3_Y - 0.09^* FL1_{DY} + FL1_{DY}}$, where $FL3_{DY}$ is the emission at YO3-biotin wavelength after DFHBI-1T excitation with both DFHBI-1T and YO3-biotin present, $FL3_Y$ is the emission at YO3-biotin wavelength after DFHBI-1T excitation with only YO3-biotin present, and $FL1_{DY}$ is the emission at DFHBI-1T wavelength after DFHBI-1T excitation with both DFHBI-1T and YO3-biotin present.

**Fluorescence microscopy.** Fluorescence microscopy images were acquired using a Zeiss LSM 800 using the 100X oil objective. The bacteria were immobilized on poly-L-lysine (PLL) coated 96 well plates (μ-Plate, uncoated, ibidi). 200 μl PLL (10 mg/ml) was applied to the wells for 30 min at 37 °C before washed with nuclease free $H_2O$ and dried. 50 μl culture prepared as described above in 1X M9 salts containing 100 μM DFHBI-1T and 17 μM YO3-biotin was added to the wells. DFHBI-1T was excited using the 488 nm laser and recorded using EGPF filter, and YO3-biotin was excited using the 555 nm laser and recorded using Texas Red filter. FRET signal was obtained by exciting using the 488 nm laser and recording using the Texas red filter. The recorded images were processed in MatLab (the code is available from our webpage www.andersen-lab.dk), where the DFHBI-1T channel was used to locate individual cells and define their boundaries. The average intensity per pixel within each cell was calculated for all three channels. Cells were selected using the directly excited YO3-biotin fluorescence and cells with the bottom 5% intensity, which corresponds to cells showing no or very low YO3-biotin fluorescence were discarded. The values from the FRET and DFHBI-1T channels of the remaining cells were used to calculate the relative FRET outputs using the equation $FRET = \frac{I_{DY}(ex_D, em_Y)}{I_{DY}(ex_D, em_Y) + I_{DY}(ex_D, em_D)}$, where $I_{DY}(ex_D, em_Y)$ is the emission at YO3-biotin wavelength (Texas Red filter) after DFHBI-1T excitation and $I_{DY}(ex_D, em_D)$ is the emission at DFHBI-1T wavelength (EGFP filter) after DFHBI-1T excitation. Both were measured with DFHBI-1T and YO3-biotin present. A FRET heatmap was created by calculating the relative FRET output per pixel within the boundaries of the selected cells by using the same equation.

**Data availability.** The authors declare that the main data supporting the findings of this study are available within the article and its Supplementary Information file. Extra data are available from the corresponding author upon request.

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

## Acknowledgements

Thanks to Rita Rosendahl and Claus Bus for technical assistance. The branched kissing-loop motif was inspired by the Z-motif developed by Di Liu and Yossi Weissman. Thanks to Daniel Svane for writing a cell sorting program for analysis of the confocal microscopy images. Thanks to Ilenia Manuguerra for helping setting up the flow cytometer for bacterial measurements. The project was funded by the Danish Council for Independent Research (Grant Numbers 0602-01772B and 4181-00573A), the European Research Council (Grant Number 683305) and the Danish National Research Foundation (DNRF81). Thanks to Cody Geary for guidance and helpful discussions concerning RNA origami design. Thanks to Cody Geary and Paul Rothemund for valuable comments on the manuscript.

## Author contributions

M.D.E.J. and S.M.S. contributed equally to the work, performed the fluorometric measurements and analyses, and made 3D and sequence designs. G.G. calculated the dipole moment. T.B.N. synthesized the YO3-biotin fluorophore. A.H.L. and M.D.E.J. performed the in vivo experiments. M.D.E.J., S.M.S., and E.S.A. wrote the manuscript with assistance from all other authors. The project was conceived by E.S.A. and supervised by K.V.G. and E.S.A.
