## [Peer Review File · Nature Communications]

REVIEWERS' COMMENTS:

Reviewer #1 (Remarks to the Author):

Jepsen et. al. describe the development of a FRET system using two different fluorescent RNA aptamers in an RNA origami scaffold. First, they made a new dye molecule, YO3-biotin, and showed that it bound the Mango aptamer in place of the published TO3-biotin dye – the benefit of YO3-biotin is that its spectral overlap with DFHBI-1T enables better FRET than TO3-biotin. Using the RNA tile as scaffold, they can tune FRET output as a function of stem length. They then show that they can design FRET systems that sense conformational changes brought about by miRNA binding (via toehold displacement disrupting the kissing loop) or SAM ligand binding (via SAM riboswitch fused to the Mango aptamer). Finally, they express apta-FRET constructs (constitutive, not the sensors) and show that they give FRET signals in live *E. coli*.

What is original: the YO3-biotin dye-Mango aptamer pairing, showing that it can undergo FRET energy transfer with DFHBI-1T-Spinach aptamer pairing. Making FRET biosensors with RNA tiles and these two dye-aptamer pairs.

Interest to community: the apta-FRET system likely will be of interest to the broader nucleic acid community, especially those working on DNA/RNA nanotechnology, biosensors, dye-binding aptamers; and to synthetic biologists. The sensing work is proof-of-principle only and in vitro, so likely will not impress biologists or others interested in detecting miRNAs or SAM.

The revised work is well written and the data are convincing to support the conclusions. It is a nice solid study, subjectively I view it on the level of a very good Nucleic Acids Res paper (top journal in the field). However, as a proof-of-principle study, it does not have the potential impact on other fields that I would expect, but I may be mis-calibrated as to the scope of this journal.

Minor comment: Supplemental data (including figure legends) and Fig. 4 describe results for Broccoli constructs, but there is no mention in main text.

Reviewer #2 (Remarks to the Author):

I think the revised manuscript is suitable for publication. I note that, to be scholarly, and since (as is appropriate since the atomic coordinates were used for modeling) the Mango crystal structure (Trachman et al, 2017) is being cited, the Spinach structures (there were two contemporaneous publications (Huang et al., 2014, Warner et al., 2014) should also be cited.

Reviewer #3 (Remarks to the Author):

In this manuscript, Jepsen and coworkers develop a genetically encodable FRET system using previously reported fluorescent RNA aptamer fluorophore complexes, Spinach and RNA Mango constructed on an RNA origami scaffold. For the better spectral overlap between the fluorophores and FRET signal, they synthesized a new fluorophore (YO3) that binds to RNA Mango aptamer. The FRET device is further fine-tuned by the orientation of dipole moment and optimizing the flexibility constraints of the two aptamers by increasing or decreasing the basepairs from the common crossover point. The applicability of the apta-FRET device for observing conformational changes in the RNA is demonstrated using microRNA invader induced conformational change and binding of the metabolite S-adenosylmethionine (SAM) to a sensing domain in the origami scaffold. Furthermore, the genetic encodability of this aptamer-based FRET system is verified by expressing it into the bacterial cells and showing that the aptamers in the origami scaffold fold correctly even inside the cells.

Overall, the manuscript has been improved significantly from the initial submission. The data presented are sound and support the conclusions. Although previous reviewers were skeptical about the novelty of the work, I agree with the authors' claim that this is the first RNA aptamer-based FRET system expressed in the bacterial cells and could be used for a variety of applications involving monitoring the conformational and folding changes of the RNAs in vivo. The authors have addressed the previous reviewers comments and criticisms satisfactorily. In my opinion, the manuscript can be accepted for publication in Nature Communication with a minor change.

Comments

The authors should give a citation for the Spinach structure after the first sentence in the 'optimization of the FRET efficiency' section.

Response to referee comments:

Below we give our response to the comments by the referees marked in red.

Reviewer #1

Jepsen et. al. describe the development of a FRET system using two different fluorescent RNA aptamers in an RNA origami scaffold. First, they made a new dye molecule, YO3-biotin, and showed that it bound the Mango aptamer in place of the published TO3-biotin dye – the benefit of YO3-biotin is that its spectral overlap with DFHBI-1T enables better FRET than TO3-biotin. Using the RNA tile as scaffold, they can tune FRET output as a function of stem length. They then show that they can design FRET systems that sense conformational changes brought about by miRNA binding (via toehold displacement disrupting the kissing loop) or SAM ligand binding (via SAM riboswitch fused to the Mango aptamer). Finally, they express apta-FRET constructs (constitutive, not the sensors) and show that they give FRET signals in live *E. coli*.

What is original: the YO3-biotin dye-Mango aptamer pairing, showing that it can undergo FRET energy transfer with DFHBI-1T-Spinach aptamer pairing. Making FRET biosensors with RNA tiles and these two dye-aptamer pairs.

Interest to community: the apta-FRET system likely will be of interest to the broader nucleic acid community, especially those working on DNA/RNA nanotechnology, biosensors, dye-binding aptamers; and to synthetic biologists. The sensing work is proof-of-principle only and *in vitro*, so likely will not impress biologists or others interested in detecting miRNAs or SAM.

The revised work is well written and the data are convincing to support the conclusions. It is a nice solid study, subjectively I view it on the level of a very good Nucleic Acids Res paper (top journal in the field). However, as a proof-of-principle study, it does not have the potential impact on other fields that I would expect, but I may be mis-calibrated as to the scope of this journal.

Minor comment: Supplemental data (including figure legends) and Fig. 4 describe results for Broccoli constructs, but there is no mention in main text.

Response:

We thank the reviewer for nice comments and agree that the current paper is mainly proof-of-principle and that future studies should aim at demonstrating sensing *in vivo*. However, we feel that the proof-of-principle has big impact because of the many applications that the aptamer-FRET can be used for and thus think that it is important to publish our results at this stage.

We agree that the paper could have been fitting for Nucleic Acid Research, but thought that the possibility to make RNA nanostructures in cells may be of interest to a wider interdisciplinary readership and may inspire them to study the system from a biophysical point of view or a more application oriented point of view.

Concerning the minor comment, the Broccoli construct is actually mentioned several places in the main text (bottom of page 3 and bottom of page 5) and references are made to all supplementary figures (4, 5, 8 and 9) showing experiments involving Broccoli.

We do however realize that the argument for using Broccoli only was described at the bottom of page 3, while it is not argued well at bottom of page 5, and have therefore elaborated with: "Both S*5-M5 and B*5-M5 were tested to compare FRET outputs of these constructs since Broccoli is optimized for *in vivo* work, but constructs containing Spinach gave higher FRET outputs *in vitro*."

Reviewer #2

I think the revised manuscript is suitable for publication. I note that, to be scholarly, and since (as is appropriate since the atomic coordinates were used for modeling) the Mango crystal structure (Trachman et al, 2017) is being cited, the Spinach structures (there were two contemporaneous publications (Huang et al., 2014, Warner et al., 2014) should also be cited.

Response:

We thank the reviewer for finding our manuscript ready for publication, and have added the suggested publications in our introduction.

Reviewer #3

In this manuscript, Jepsen and coworkers develop a genetically encodable FRET system using previously reported fluorescent RNA aptamer fluorophore complexes, Spinach and RNA Mango constructed on an RNA origami scaffold. For the better spectral overlap between the fluorophores and FRET signal, they synthesized a new fluorophore (YO3) that binds to RNA Mango aptamer. The FRET device is further fine-tuned by the orientation of dipole moment and optimizing the flexibility constraints of the two aptamers by increasing or decreasing the basepairs from the common crossover point. The applicability of the apta-FRET device for observing conformational changes in the RNA is demonstrated using microRNA invader induced conformational change and binding of the metabolite S-adenosylmethionine (SAM) to a sensing domain in the origami scaffold. Furthermore, the genetic encodability of this aptamer-based FRET system is verified by expressing it into the bacterial cells and showing that the aptamers in the origami scaffold fold correctly even inside the cells.

Overall, the manuscript has been improved significantly from the initial submission. The data presented are sound and support the conclusions. Although previous reviewers were skeptical about the novelty of the work, I agree with the authors' claim that this is the first RNA aptamer-based FRET system expressed in the bacterial cells and could be used for a variety of applications involving monitoring the conformational and folding changes of the RNAs in vivo. The authors have addressed the previous reviewers comments and criticisms satisfactorily. In my opinion, the manuscript can be accepted for publication in Nature Communication with a minor change.

Comments

The authors should give a citation for the Spinach structure after the first sentence in the 'optimization of the FRET efficiency' section.

Response:

We thank the reviewer for the nice comments, and appreciate that the possible applications of our apta-FRET system are recognized by the reviewer. We also thank for the observation that we had missed citations for the Spinach structure, which has now been added at the suggested position in the text.